# A Disposable Electromagnetic Bi-Directional Micropump Utilizing a Rotating Multi-Pole Ring Magnetic Coupling

**DOI:** 10.3390/mi13101565

**Published:** 2022-09-21

**Authors:** Chao Qi, Naohiro Sugita, Tadahiko Shinshi

**Affiliations:** 1Department of Mechanical Engineering, Tokyo Institute of Technology, 4259 Nagatsuta-cho, Midori-ku, Yokohama 226-8503, Japan; 2Laboratory for Future Interdisciplinary Research of Science and Technology (FIRST), Institute of Innovative Research (IIR), Tokyo Institute of Technology, 4259 Nagatsuta-cho, Midori-ku, Yokohama 226-8503, Japan

**Keywords:** disposable micropump, discrete peristaltic micropump, bi-directional pumping, high pump density, high-throughput rapid PCR, magnetic coupling

## Abstract

Electromagnetic bi-directional micropumps (EMBM) are indispensable for the development of portable devices which enable fluid transportation in forward and reverse directions. However, the high disposal cost of rare-earth magnets attached to the fluidic part and the low pump density due to multiple motors limit their practical application in disposable multi-channel microfluidic applications such as droplet-based oscillatory-flow (DBOF) rapid PCR amplification. Therefore, this paper presented a low-cost, disposable, high-pump-density EMBM. To reduce the disposal cost, we separated the magnets from the disposable fluidic part and used cylindrical holes to store and guide the magnet, which resulted in the ability to reuse all the magnets. To increase the pump density, we used the combination of one motor and one multi-pole ring magnet to drive several channels simultaneously. A proof-of-concept prototype with a pump density of 0.28 cm^−2^ was fabricated and experimentally evaluated. The fabricated micropump exhibited a maximum flow rate of 0.86 mL/min and a maximum backpressure of 0.5 kPa at a resonant frequency around 50 Hz. The developed multi-channel micropump with reusable magnets is highly beneficial to the development of low-cost and high-throughput rapid PCR amplification microchips and therefore can have a significant impact on timely infectious disease recognition and intervention.

## 1. Introduction

The polymerase chain reaction (PCR) is a powerful and simple molecular biology technique that can amplify in-vitro DNA fragments of interest. It is widely used in disease diagnosis, DNA sequencing, and forensics [1,2,3]. The most commercially available PCR instruments exponentially amplify via a repetitive multi-step thermal cycle. However, the large thermal inertia of these bulky instruments constrains rapid temperature changes of the sample, which usually leads to a reaction time of several hours. The long reaction time and bulky size are major disadvantages when applying this technique to emerging infectious diseases. Rapid point-of-care (POC) PCR diagnoses can significantly enhance timely disease recognition and intervention [4].

The microfluidic PCR amplification chip, developed with the advent of MEMS technology, thus attracts considerable attention because of its fast reaction capability, its portability, and its low reagent consumption. Amongst microfluidic PCR amplification chips, many efforts have been made based on the droplet-based oscillatory-flow (DBOF) technique since this can be integrated with pre-processing and post-processing within the PCR diagnostic equipment to realize a low-cost and fully automated micro total analysis system (μTAS). Such systems have been reported to have fast reaction times down to 13 min [5].

The DBOF-PCR amplification chip rapidly achieves thermal cycling by moving the droplet containing the PCR mixture back and forth amongst three temperature regions. As a result, a rapid temperature change of the PCR mixture is realized due to the small droplet volume. However, because a bulky syringe pump with low pump density (channel number/pump footprint) has been typically used to drive the working fluid [5,6,7], issues of non-portability and low throughput of the DBOF-PCR system have limited its practical application. To develop a POC high-throughput DBOF-PCR amplification system, it is necessary to develop a portable bi-directional micropump with a high pump density.

Various bi-directional micropumps have been developed based on different actuation mechanisms such as piezoelectric actuation [8], pneumatic actuation [9], and electrostatic actuation [10]. Electromagnetically driven micropumps are favorable for portable medical devices thanks to their low drive voltage and environmental insensitivity compared with the micropumps mentioned above. Electromagnetic bi-directional micropumps previously developed are mainly divided into two categories: (1) electromagnetic direct drives; (2) motor and magnetic transmission mechanisms.

For the direct electromagnetic drive approach, Amrani [11], Chee [12], and Rusli [13] developed a valveless rectification pump using an electromagnetic membrane actuator (EMMA) consisting of a bulky coil, a flexible membrane, and a permanent magnet (PM) to drive the fluid. Bi-directional pumping was achieved using a pair of valveless rectification pumps. However, using two bulky coils lowers the pump density and increases the power consumption (0.5~1 W).

To reduce the power consumption, micropumps using a motor and a magnetic transmission mechanism such as magnetic coupling and magnetic fluid have been proposed [14,15,16,17,18]. In these micropumps, the direction of the fluid was controlled by the rotation direction of the motor. Kim [14] used several rotating PMs driven by a motor to control magnetic fluid droplets so that they could push the working fluid. However, the magnetic fluid could not maintain its shape at 15 rpm, resulting in a low flow rate (3.8 μL/min) below the DBOF-PCR requirement (from 0.03 to 0.11 mL/min [6,7]). Du [15] and Yobas [16] used several PMs rotated by one motor to control several steel balls to pump the working liquid. Although the pump performance was enhanced (1 and 5 mL/min), the pump showed poor reliability since the large centrifugal force would easily pull the steel balls off the pump. Pan [17] and Shen [18] proposed a micropump with three discrete reciprocating membranes to enhance the stability. Each membrane was bonded to a PM. The reciprocating motion of one membrane/magnet was realized by rotating another PM below the previous PM and mounted on a motor shaft. The unidirectional and bi-directional pumping were achieved by arranging three lower PMs mounted on the shaft with a 2π/3 phase difference and changing the motor rotation direction, respectively.

Any part of the micropump contacting the PCR mixture must be discarded to avoid cross-contamination of PCR samples after use. If the micropumps mentioned above using integrated magnets [17,18] were applied to high-throughput PCR tests, many expensive rare-earth PMs would need to be discarded. In addition to their high running cost, a large motor is used to drive fluid in one channel, significantly increasing the pump density (0.1 cm^−2^ in ref. [18]). Consequently, in a limited footprint, a cost-effective high-throughput DBOF-PCR cannot be realized via these approaches.

Therefore, this research aimed to develop a disposable electromagnetic bi-directional micropump with a high pump density and no disposable permanent magnet elements. We propose to use a multi-pole ring magnet connected to the motor to drive small magnets stored in the corresponding cylindrical hole under the pump membrane. Since the membrane side magnets are all stored in the cylindrical holes instead of being integrated with the membrane as in previous literature [17,18], these can be reused afterwards for further tests. By increasing the pole number of the motor side ring magnet and still using just one motor, multi-channel pumping can be realized such that the pump density can be enhanced.

In this paper, the actuator force, membrane displacements, and the pump performance were demonstrated after first introducing the design and fabrication of the proof-of-concept two-channel micropump. For the future design of a micropump with more channels, it is necessary to develop an effective model of the micropump. Therefore, a dynamic and fluid model of the developed pump was built through a lumped-element dynamic analysis model of the small magnet directly driving the membrane and fluid–structure interaction analysis model to confirm that the prototype pump performance was the same as expected. The simulated results were compared with those simulated using conventional equivalent electric circuits [18,19].

## 2. Proposal of a Disposable High-Density Bi-Directional Electromagnetic Micropump

The schematic of the proposed electromagnetic micropump to increase the pump density and eliminate PM disposal is shown in Figure 1a,b. The micropump consists of a channel layer, a flexible membrane, three cylindrical magnets, a retainer layer with three cylindrical holes, and an axially magnetized two-pole ring magnet driven by a motor. All the cylindrical magnets were loaded into corresponding cylindrical holes. Each cylindrical magnet experiences an alternating magnetic force caused by the ring magnet rotation. While the force is repulsive, the cylindrical magnet will push the membrane upwards, increasing the fluid pressure in the corresponding chamber. While the force is attractive, the magnet will return to the cylindrical hole, and the membrane will return to equilibrium due to its elastic restoring force.

Three magnets were spaced equally along a circle, and a six-phase actuation sequence, as shown in Figure 1c, is applied to pump the working fluid in one direction. It is only necessary to change the motor rotation direction to change the pumping direction. With this configuration, since all the cylindrical magnets are separated from the membrane, only fluidic parts made of polymer will need to be disposed, decreasing the running cost. In addition, with this approach, the pump density can be increased by increasing the number of channels, ring magnet poles, and cylindrical magnets.

## 3. Pump Design and Fabrication

### 3.1. Configuration of a Large-Scale Micropump

To verify the feasibility of the proposed micropump, we refer to our previous work which discussed the development of a large-scale model of a single-channel micropump and its basic experimental evaluation [20]. In this work, a large-scale model of a two-channel micropump with a footprint of 225π mm^2^ utilizing a four-pole ring NdFeB magnet with dimensions of Φ_out_23 mm × Φ_in_20 mm × t4.7 mm (N40, NeoMag Co., Ltd., Tokyo, Japan) and a commercial brushless DC motor (EC45 flat, Maxon Motor Inc., Sachseln, Switzerland) were developed and evaluated by simulation and by experiment. Details of the model are shown in Figure 2.

The rest of the parts of the pump are a channel layer with two channels, a flexible membrane with the thickness of 0.275 mm, a spacer with a thickness of 0.3 mm, six axially magnetized cylindrical NdFeB magnets with dimensions of Φ1.43 mm × t1.47 mm (N40, NeoMag Co., Ltd., Tokyo, Japan), and a retainer layer with six cylindrical holes. Each cylindrical hole has a depth of 1.55 mm and a diameter of 1.5 mm. The diameter of the cylindrical hole is larger than that of the cylindrical magnet for a bearing gap. A spacer was used to prevent the membrane from adhering to the retainer layer. The movable circular part of the membrane in each chamber has a diameter of 6 mm. The depth of each chamber and the channel is 0.7 mm. The channel width is 1 mm. The inner diameter and height of the ports are 2 mm and 8 mm, respectively.

### 3.2. Magnet Coupling Simulation

We studied the effect of the center-to-center distance (d_c-c_) and the gap distance (g_m_) between the cylindrical and ring magnets on the generated magnetic force. The magnetostatic analysis was conducted using Maxwell 3D (19, Ansys Inc., Canonsburg, PA, USA). Figure 3 shows the analytical model. The dimensions of these magnets are the same as those mentioned in the last section. Two magnets have the same remanence B_r_ of 1.25 T and the same coercivity H_cb_ of 876 kA/m. In the simulations, we varied the d_c-c_ from 10.75 mm to 12.25 mm and the g_m_ from 1 mm to 4 mm.

Figure 4a shows the relationship between the simulated magnetostatic forces on the cylindrical magnet and the rotation angle at different d_c-c_ when g_m_ is 1 mm. The vertical force showed an alternating quasi-square waveform because the arc length of each pole of the lower ring magnet was larger than that of the diameter of the upper cylindrical magnet. The average vertical force reached the largest when d_c-c_ is 11.5 mm, equal to the mean radius of the ring magnet. The in-plane force was greater than zero during the rotation of the ring magnet since there was an eccentricity between the cylindrical magnet and ring magnet. The non-zero in-plane force generated static or kinetic friction between the cylindrical magnet and the cylindrical hole wall. The average in-plane force was the lowest when d_c-c_ was 11.5 mm. Since a large driving force and a small friction force are preferred to generate a large pump stroke, d_c-c_ was set to 11.5 mm in the following context.

Figure 4b shows the relationship between the simulated magnetostatic forces on the cylindrical magnet and the rotation angle with different g_m_. The result showed that, as g_m_ decreased, both the amplitude of the vertical and the in-plane forces increased. To determine the gap for the micropump, we simulated the static displacement of the membrane using the acquired forces at a different g_m_ in the following section.

### 3.3. Membrane Static Displacement Simulation

Since the membrane actuator acts not only as the pumping unit here, but also as the valving unit, the displacement of the membrane is critical. The height of the chamber was 0.7 mm, so the membrane displacement needs to be larger than this for an effective valving effect. Here we simulated the static displacement of the membrane using the simulative force obtained at different g_m_. The analysis model is shown in Figure 5a. The simulations were implemented in Solidworks Simulation (Ver. 2018, Solidworks, Ile-De-France, France). The Young’s modulus, Poisson ratio, and density of the PDMS membrane were 2 MPa, 0.45, and 1020 kg/m^3^, respectively. The membrane experiences a concentrical force over a Φ1.5 mm circular area. The side of the membrane was fixed in the simulation. We used the force when the rotation angle is 45° at different gaps as the constant concentrical force shown in Figure 5b. The simulation result showed that when g_m_ was smaller than 3 mm, the static displacement was larger than 0.7 mm, as shown in Figure 5c. However, since the actual magnetic force decreases as the membrane displacement increases, there is a spacer between the cylindrical magnet and membrane, and we chose 1 mm as the gap between the cylindrical magnet and ring magnet for a better valving effect.

### 3.4. Fabrication

Based on the above simulation results, the d_c-c_ and the g_m_ were set to 11.5 mm and 1 mm, respectively. We chose PMMA and PDMS as the channel layer and the flexible membrane materials, respectively, due to their biocompatibility and low cost. MEMS technology can realize the fabrication of these two layers to lower the costs [21,22]. We used an end-milling machine (monoFab SRM-20, Roland Corp., Shizuoka, Japan) to manufacture the PMMA channel layer.

A PMMA master mold machined by end milling was used for the PDMS membrane to fabricate the membrane via molding (80 °C for two hours). The PDMS pre-polymer base and its curing agent (SIM 260 and CAT 260, Shin-Etsu Chemical Co., Ltd., Tokyo, Japan) were used in a weight ratio of 10:1. To seal the PMMA channel with the PDMS flexible membrane, thermal curing of liquid PDMS for permanent bonding was used, as described in Figure 6a [23].

The adhesive PDMS layer exists only at the contact area of two surfaces and has a negligible influence on the microfluidic channel. (1) A liquid PDMS glue layer (the weight ratio of PDMS prepolymer and curing agent is 10:1) with a thickness of 30 μm was spun onto a flat PMMA surface by a spin coater (1H-DX2, MIKASA Co., Ltd., Nara, Japan). (2) The glue layer was then transferred to a flexible silicone rubber sheet (6-612-01, AS ONE, Co., Ltd., Osaka, Japan) by placing the rubber on the coated PMMA. (3) The silicone rubber sheet was peeled off the PMMA layer. (4) Subsequently, the machined PMMA channel layer was stamped on the coated rubber to transfer the glue layer. (5) The use of flexible rubber makes peeling the coated layer off the channel layer easier. (6) Afterward, the coated channel layer was placed on the flexible PDMS membrane. (7) After curing at 80 °C for two hours, the permanent sealing of the channel layer was achieved. Eventually, the PMMA connectors were glued to the micropump with epoxy (RT30, Araldite, The Woodlands, TX, USA) at room temperature for one hour.

The micropump parts are illustrated in Figure 6b. For the spacer with six Φ6 mm holes and the retainer layer with six Φ1.5 mm × t1.55 mm cylindrical holes, we chose duralumin (A2017) as the material since it is not magnetic. The 0.3 mm duralumin spacer and retainer layer were machined using end milling. The holder of the ring magnet was fabricated by a Stereolithography Apparatus (SLA) 3D printer (Form 3, Formlabs, Somerville, MA, USA) utilizing resin (clear resin, Formlabs, Somerville, MA, USA). All the layers were clamped using polymer wing screws. The prototype of the two-channel large-model micropump is illustrated in Figure 6c.

## 4. Experimental Evaluation

### 4.1. Measurement of the Static Magnetic Force

Before measuring the static magnetic force, we measured the magnetic flux density to evaluate the actual magnetic properties. A custom-made magnetic flux sensor using Hall element (HG-0711, Asahi Kasei Microdevices, Tokyo, Japan) was used to scan the centerlines 100 μm above the top surface of the cylindrical magnet and ring magnet, respectively, as shown in Figure 7a. The measured magnetic flux density in the Z-axis is shown in Figure 7b. The measured magnetic flux density in the Z-axis of the cylindrical magnet was 80% of the simulated value.

The measurement setup for the static force is shown in Figure 8a. The upper magnet was positioned by a custom-made six-DOF micro-positioner with a resolution of 10 μm. In this experiment, the ring magnet was mounted on the holder, which was rotated by the motor (KeiganMotor KM-1U, Keigan Co., Ltd., Kyoto, Japan) at a rotation speed of 0.1 Hz. The d_c-c_ distance was set to 11.5 mm, and the g_m_ varied from 1 to 4 mm. When the motor started to rotate, we could indirectly obtain the force on the cylindrical magnet through a digital balance (GF-8000, A&D) with a sampling capability of 10 Hz. The measurement results and the simulation results are shown in Figure 8b. It should be noted that the B_r_ and H_cb_ of the cylindrical magnet used in simulations were modified to 80% of the catalog values based on the magnetic flux density measurement results. The measured forces had a good agreement with the simulated ones.

### 4.2. Measurement and Modelling of the Membrane Dynamic Response

#### 4.2.1. Setups for Measuring the Membrane Dynamic Response and Identifying Its Model Parameters

The dynamic response of the membrane system is very important to the final pump performance. Here, to understand the dynamic behavior of the membrane, we measured the membrane displacement at different actuation frequency, and built a model of the membrane system when the working fluid did not flow while rotating the ring magnet.

The measurement setup of the dynamic membrane displacement is shown in Figure 9a. The four-pole ring magnet was used to drive the cylindrical magnet. A laser displacement sensor (LK-G80, Keyence, Osaka, Japan) was used to measure the displacement at the center of the membrane. The center area of the upper surface of the transparent membrane was colored light red to reflect the laser beam. By changing the motor rotational speed from 2.5 Hz to 25 Hz during the measurement, the actuation frequency of the cylindrical magnet was changed from 5 Hz to 50 Hz. In particular, at 5 Hz, the membrane displacement with and without the spacer were compared to study the influence of the spacer.

To build the membrane system model, we regarded the membrane mechanism as a one-degree-of-freedom mass-damping-spring system and experimentally identified each parameter.

The measurement setup for the spring stiffness is shown in Figure 10a. A cylindrical magnet fixed on a load cell (LTS-50GA, Kyowa Electronic Instruments, Tokyo, Japan) was pushed up by a Z-direction micro-positioner to measure the stiffness of the membrane.

Since we could not measure the natural frequency of the membrane based on the experiments, the equivalent mass was identified from the theoretical natural angular frequency of the membrane and the measured stiffness of the membrane [24,25]:
(1)$$\omega_{n}=\sqrt{\frac{k}{m}}=1.015^{2}\frac{\pi^{2}}{r^{2}}\sqrt{\frac{Et^{3}/12\left(1-v^{2}\right)}{\rho t}},$$
where *ω_n_* is the resonant frequency, *k* is the measured membrane stiffness, *m* is the equivalent mass in this mass-spring-damper system. *r* and *t* are the radius and thickness of the membrane, respectively. *E*, *ν* and *ρ* are Young’s modulus, Poisson’s ratio, and density of the PDMS, respectively.

Since the Young’s modulus of the PDMS is not constant [26], we calculated the Young’s modulus by averaging the previously measured spring constant and the circular membrane displacement equation [27]:
(2)$$E=\frac{12k\left(a^{2}ln\frac{a}{r}+\frac{r^{2}-a^{2}}{2}\right)\left(1-\nu^{2}\right)}{8\pi t^{3}},$$
where *a* is the radius of the concentrical load area.

The damping coefficient was identified by measuring and curve-fitting the transient response of the membrane. The experimental setup for measuring the transient response is shown in Figure 10b. A displacement laser sensor measured the center displacement of the Φ6 mm membrane clamped by the jigs after removing the cylindrical magnet pushing the membrane. To obtain the damping coefficient, the experimental response was fitted with the following overdamping response equation using the least-squares method:(3)$$z\left(t\right)=e^{-\zeta \omega_{n}t}\left(\frac{\zeta \omega_{n}z_{0}+v_{0}}{\sqrt{\zeta^{2}-1}\omega_{n}}sinh\sqrt{\zeta^{2}-1}\omega_{n}t+z_{0}cosh\sqrt{\zeta^{2}-1}\omega_{n}t\right),$$
where *z(t)* is the displacement function, *ζ* is the damping ratio, *t* is the time, *z_0_* is the initial displacement, and *v*_0_ is the initial velocity. The *ζ* can be calculated by:(4)$$\zeta =c/2m\omega_{n},$$
where *c* is the damping coefficient.

Here, *r*, *ρ*, *t*, *ν*, *a*, *z*_0_
*and x*_0_ are equal to 3 mm, 1020 kg/m^3^, 0.275 mm, 0.45, 0.75 mm, 0.59 mm, and 0 m/s, respectively.

#### 4.2.2. Measurement Results of the Membrane Dynamic Response and Model Parameters

The measurement results of vertical displacement at different actuation frequency as shown in Figure 9b indicate that the membrane went upwards faster than downwards within one period. Furthermore, as the actuation frequency increased, the membrane displacement decreased and became farther than the equilibrium position (Z = 0 mm). Besides, without a spacer of 0.3 mm, as shown in Figure 9b, the membrane has no displacement, which results from the stickiness of the PDMS membrane to the retainer layer top surface. The relationship between the amplitude of the membrane and actuation frequency is shown in Figure 9c. The amplitude decreased as the actuation frequency increased.

Figure 10c shows the equivalent membrane spring constant identification result. The nonlinear relationship between the force measured by the load cell and the displacement measured by the laser displacement sensor (LK-G80, Keyence, Osaka, Japan), similar to previous research about PDMS stiffness [26]. We obtained the average stiffness by linearizing the nonlinear stiffness within a moving range of 0.5 mm since the measured membrane displacement is not larger than 0.5 mm. The linearized spring constant was 39.4 N/m. The calculated equivalent mass was thus equal to 1.61 × 10^−6^ kg based on equation (1) and the linearized spring constant. The measured result and the fitting curves of the membrane transient response are shown in Figure 10d. According to the curve fitting, the damping coefficient was identified as 0.3425 kg/s. Hence the damping ratio *ζ* of the system was equal to 21, which means the system is overdamping. Besides, the force from the cylindrical magnet to the membrane was periodical. The large damping ratio (>1) of the membrane along with the periodical force exactly explains why we could not measure the natural frequency and the relationship between the amplitude and actuation frequency in Figure 9c. Meanwhile, we can also know that when we drive at a frequency no smaller than 5 Hz, we will obtain the maximum displacement of the membrane at 5 Hz.

### 4.3. Measurement of the Magnet Dynamic Response

#### 4.3.1. Setup for Measuring the Magnet Dynamic Response

The cylindrical magnet is the most essential component of the pump since it transmits the rotation into the desired vertical vibration to drive the flow. However, we found the cylindrical magnets were wobbling during operation. Because the force on the cylindrical magnet is affected by its tilting angle, we studied the motion of the cylindrical magnet.

The measurement setup of magnet vertical displacement and its tilting angle is shown in Figure 11a. The four-pole ring magnet was used to drive the cylindrical magnet. We measured the displacements at two locations of the cylindrical magnet. One is the center, and the other is 300 μm away from the center, at an actuation frequency of 5 Hz. Due to the space limitation for the laser displacement sensor, simultaneous measurement of two spots was difficult. As a result, we measured the displacement at those locations individually. A Hall element sensor was used to detect the rotation angle of the ring magnet, and then we can know the displacements of two spots at the same rotation angle of the ring magnet.

#### 4.3.2. Measurement Results of the Magnet Dynamic Response

The measurement results are shown in Figure 11b. The tilting angle *θ* can be obtained from the following equation:(5)$$\theta =\left(\frac{\Delta Z\;}{0.3}\right)=\left(\frac{Z_{c}-Z_{offset}}{0.3}\right),$$
where *Z_C_* is the displacement of the membrane center, and *Z_offset_* is the displacement of location 300 μm away from the center along the Y-axis.

Figure 11b shows that the tilting angle of the cylindrical magnet alternated during the ring magnet rotation. A positive value represents the cylindrical magnet rotates around the positive X-axis, and the negative value represents the magnet rotates around the negative X-axis. The amplitude of the negative tilting angle was larger than that of the positive one. The vertical displacement was smaller when the cylindrical magnet rotated around the positive X-axis. In this case, the tilting of the cylindrical magnet was restricted by the cylindrical hole wall to a greater degree, as shown in Figure 11c. When the magnet partially leaves the cylindrical hole due to the positive vertical force, the torque around the negative X-axis will tilt the cylindrical magnet around the negative X-axis. As a result, to understand the cylindrical magnet motion, it is necessary to know the vertical force and torque around the X-axis.

The force and torque values on the cylindrical magnet are affected by the gap between the cylindrical magnet and ring magnet, the ring magnet rotation angle, and tilting angle of the cylindrical magnet. We already simulated the effect of the gap and the rotation angle on the forces of the cylindrical magnets, as shown in Figure 4b. Here we simulated the effect of the tilting angle of the cylindrical magnet on the magnetostatic forces and the torque when the vertical gap between the cylindrical magnet and ring magnet is equal to 1 mm. The analysis model is shown in Figure 12a. In simulations, we varied the tilting angle α_t_ of the cylindrical magnet around the X-axis from 0° to 20°. Based on the results in Figure 12b, we can know that the magnetic forces do not significantly change within a tilting angle of 20°. However, the torque in the X-axis becomes larger when the tilting angle becomes larger. Since the membrane system is overdamped, the large displacement of the magnet/membrane happens when the actuation frequency is low. When the displacement becomes larger, the titling angle also becomes larger since the cylindrical hole wall will restrict it to a lower degree. Thus, from 5 Hz to a higher rotation frequency, the maximum amplitude of the tilting angle is no greater than 7.5°. Based on the simulation results in Figure 12b, for the designed micropump, we know that the torque around the X-axis was affected by the tilting angle. However, the vertical force and in-plane force were only slightly affected by the tilting angle when the tilting angle was smaller than 20°. Based on the measurements, for the developed micropump, the tilting angle of the cylindrical magnet had a relatively insignificant effect on the forces on the cylindrical magnet itself.

Based on the measured rotation angle of the ring magnet, the measured displacement of the cylindrical magnet and the calculated titling angle are shown in Figure 11b, combined with the simulated results in Figure 4a and Figure 12b, we obtain the simulated vertical force and torque around X-axis as shown in Figure 11b. The vertical displacement of the magnet is controlled by the vertical force. We found that there was a phase delay between the vertical force and the displacement, which confirms the existence of the friction. Friction has a negative effect on pump performance because it not only reduces the driving force, but also causes wear of the cylindrical magnets. Therefore, it is important to reduce friction to achieve a high performance and to run the pump for a long time.

### 4.4. Pump Performance

#### 4.4.1. Flow Rate, Pressure, and Pump Temperature during Continuous Operation

We evaluated the pump performance at different actuation frequencies in the forward and backward directions. The actuation frequency was changed by tuning the rotation speed of the motor, and the motor rotation direction changes the pumping direction. The experimental setups for measuring the maximum averaged flow rate, the maximum averaged backpressure, and the measured results are shown in Figure 13.

Both the forward and backward pumping showed the same relationship between the actuation frequency and the maximum averaged flow rate and backpressure. For the forward pumping direction, the maximum flow rate reached a peak of 0.78 mL/min at a frequency of 50 Hz, and the maximum backpressure reached a peak of 0.4 kPa at 50 Hz. For the backward pumping direction, the maximum flow rate reached a peak of 0.86 mL/min at 60 Hz, and the maximum backpressure reached 0.5 kPa at 50 Hz.

After one hour of continuous operation at a rotation frequency of 50 Hz without any backpressure, the pump temperature measured by an infrared camera (FLIR E4 Wi-Fi, FLIR Corp., Wilsonville, OR, USA) is shown in Figure 13c. The result shows that the temperature did not change significantly. Additionally, the measured pump performance was almost the same as the one at the beginning, which means the PDMS membrane is durable for one-hour operation. As a result, we believe the developed micropump could be used for PCR amplification in continuous operation.

#### 4.4.2. Applicability of the Prototype Pump to Point-of-Care PCR

The criteria for selecting a micropump for high-throughput point-of-care DBOF-PCR amplification are a high pump density, a low fabrication fee, a low running fee, a suitable flow rate (0.03~0.11 mL/min [6,7], and a low driving voltage as well as low power consumption. Table 1 summarizes the features of our developed micropump against previously developed bi-directional electromagnetic micropumps reported in other literature, assuming *n* channels are required. The motor input power of this study was measured using a digital power meter (WT-210, Yokogawa Corp., Tokyo, Japan) at 50 Hz.

Compared with pumps reported in the literature, the developed micropump in this study takes advantage of low running cost, high pump density (when *n* is equal to or larger than 2), and stable operation. However, since a commercial BLDC motor was used for the feasibility verification, the power consumption and driving voltage was relatively large.

## 5. Pump Modeling

### 5.1. Modeling Overview

For the future design of a micropump with more channels, it is necessary to develop a model of the micropump. Therefore, we discuss the modeling of the micropump in the next section.

Building an equivalent electric circuit of the fluidic circuit has been a commonly used approach for pump analysis [18,19]. Although this method has the advantage of simplicity, many experiments and calculations are needed to obtain the parameters for the model. Furthermore, the representation of physical processes in a lumped model is crude, which means the accuracy cannot be guaranteed.

To this end, a finite element analysis using the fluid–structure interaction (FSI) model, which couples the analysis of the transient structure to a fluid analysis, has been proposed as an alternative [28,29]. Since the membrane displacement in the pump is not negligible, a two-way interactively implicit simulation was used. The transient structural analysis system receives the force data from the motion of the liquid and then solves the structural behavior over time.

The displacement data from the motion of the plate is received by the computational fluid analysis (CFD) package (Fluent, Ansys Inc., Canonsburg, PA, USA), which then solves the fluid behavior over time. In this work, since the membrane oscillation depends on the impact of the cylindrical magnet actuated by the ring magnet, it is necessary to obtain the force. Hence, we built a simple lumped model of a magnet and membrane to obtain the force on the membrane over time.

### 5.2. Dynamic Modeling of Magnet Coupling and Membrane

A simple lumped two-mass model of one magnet and one membrane with a working fluid in one chamber is proposed, as shown in Figure 14. During the rotation of the four-pole ring magnet, the cylindrical magnet experiences a vertical magnetic force *F_z_* and an in-plane magnetic force *F_xy_* from the ring magnet. The vertical magnetic force drives the cylindrical magnet to create reciprocating motion. The in-plane force generates friction *f* dynamically depending on the motion state of the cylindrical magnet as shown in Figure 14. The gravitational force of a cylindrical magnet is *m_PM_*g. Although the cylindrical magnet will cause a tilt of the torque around the X-axis, the force is mainly dependent on the rotation angle of the ring magnet and the distance between the cylindrical magnet and ring magnet. The influence of the tilting angle on the vertical and in-plane forces can be neglected based on our experiment above in Figure 11 and the simulation results in Figure 12.

The dynamic model was numerically solved using MATLAB (R2020b, Mathworks, Natick, MA, USA). The initial positions *z(0)* and velocities *v(0)* for the magnet and the ring magnet were set to zero. When the magnet goes downward and contacts the cylindrical hole bottom, we assume that the velocity will become zero instantly. Besides this, the velocity stays at zero at this position if the acceleration is no greater than zero. When the magnet and membrane collide, we used the coefficient of restitution and theorem of momentum to obtain the velocity of the membrane and magnet at the next time point. When the absolute value of the negative velocity of the magnet is larger than that of the membrane, the magnet will detach from the membrane. The membrane will then move according to its natural frequency.

In the steady state, the force *F_membrane_* on the membrane can be obtained from the following equation:(6)$$F_{membrane}\left(t\right)=m\ddot{z\left(t\right)}+c\dot{z\left(t\right)}+kz\left(t\right),$$
where *m* is the total mass of the membrane mass (*m_PDMS_*) and the water mass (*m_water_*) in the chamber, *c* is the damping coefficient of the membrane, *k* is the spring stiffness of the membrane, and *z(t)* is the displacement of the membrane at the center.

According to our measurements, *k*, *c, m_PDMS_* + *m_water_* were 39.40 N/m, 0.3425 kg/s, and 2.11 × 10^−5^ kg, respectively. The cylindrical magnet *m_PM_* had a mass of 1.78 × 10^−5^ kg. The static friction coefficient and kinetic friction coefficient were 0.5 and 0.33, respectively. The coefficient of restitution between the cylindrical magnet and the PDMS membrane was 0.77.

### 5.3. Pump Simulation Using the Fluid–Structure Interaction (FSI) Method

The analysis model we developed using the FSI method is shown in Figure 15 for comparison with the simplified model results. We used Ansys Workbench (19.0, Ansys Inc., Canonsburg, PA, USA) as the software for the simulation. For the mechanical system, the membrane has Young’s modulus of 2.45 MPa, a Poisson’s ratio of 0.45, and a density of 1020 kg/m^3^. The forces from the cylindrical magnets on the three membranes were applied with a phase difference of 2π/3, according to the results from the previous section. The force area is a circle with a diameter of 1.5 mm on the PDMS membrane. All the sides of the membrane are completely fixed. The fluid in the simulation was water. The pressures of both ports were set to zero Pa. The boundary condition between the membrane and the fluid is the fluid structural interface.

A lumped model using an equivalent electrical circuit [18] was used to conduct a comparison. The equivalent electrical circuit parameters (resistance, capacitance, and inductance) can be found in previous research [30].

The results of the average flow rate through the outlet based on the FSI model and the equivalent electrical circuit are shown in Figure 16a,b. The simulation results based on the FSI model showed a good agreement with the measurement results. However, for the results based on the equivalent electrical circuit, only the ones at a low actuation frequency showed a good agreement. These observations are similar to previous studies [18]. To explain the above disagreement, we calculated the stroke volume of the pump per second *U(f)* based on the following equation:(7)$$U\left(f\right)=\left[V{\left(f\right)}_{max}-V{\left(f\right)}_{min}\right]f,$$
where *f* is the actuation frequency, *V(f)_max_* is the maximum membrane volume change at a frequency at *f*, and *V(f)_min_* is the minimum membrane volume change at *f*.

*V(f)_max_* and *V(f)_min_* are equal to the volume change of the circular membrane when the center displacement is maximum and minimum, respectively, under a concentrated load. The equations for membrane deformation under a concentrated load can be found in the literature [27].

The results in Figure 16c show that the stroke volume per second increased as the actuation frequency increased to 70 Hz. Afterwards, the stroke volume decreased as the frequency increased. In this type of pump, two side chambers function as the valves for the middle chamber. The final flow rate depends on the stroke per second and the valve efficiency. A large stroke per second and high valve efficiency will contribute to a high flow rate. Based on the overdamped system in Figure 9c, the membrane stroke will decrease as the actuation frequency increases, which means the valve efficiency decreases as the frequency increases. Since the pump efficiency is represented by a constant in the conventional equivalent electrical circuit model, the actual pump efficiency is lower than that expected for the equivalent electrical circuit. As a result, the flow rate in the high actuation frequency range is higher than that of measurement. We can also know that to further improve the pump performance, we can increase the membrane valve efficiency either by decreasing the membrane thickness or improving the magnetic force between the magnetic coupling. However, since the membrane experiences periodical impact force from the cylindrical magnet, it will fracture if the membrane is too thin or the force is too large. As a result, to obtain both a good valve effect and durability, the suitable membrane thickness should be studied simulatively or experimentally.

## 6. Conclusions and Outlook

In this study we proposed, fabricated, and evaluated a disposable bi-directional electromagnetic micropump with a high pump density. To reuse all the rare-earth magnets, we used cylindrical holes as a guide for the magnets. A multi-pole ring magnet was used to drive more than one channel within a single micropump to increase the pump density. The use of a multi-pole ring magnet also simplified the configuration of the micropump. Compared with a traditional electromagnetic bi-directional micropump utilizing magnets attached to the membrane, and three asynchronous magnets, this micropump has the advantage of a lower disposal fee, a higher pump density, and a simpler configuration. Using a four-pole ring magnet, we prototyped a proof-of-concept two-channel micropump with a high pump density (0.28 cm^−2^). The experimental evaluation of the micropump showed a maximum flow rate of 0.86 mL/min against zero backpressure, and a maximum backpressure of 0.5 kPa at zero flow rate at the resonant frequency of around 50 Hz. Operational testing showed that the developed micropump can continuously work for one hour without any significant temperature rise. To simulate the developed micropump, a simple lumped model was firstly used to obtain the impact on the membrane from the magnet. A fluid–structural interaction model was then used to simulate the micropump performance. Compared with the conventional equivalent electric circuit model, this combination of a simple lumped model and a fluid—structural interaction model showed a better agreement to the measurement results. We expect our developed micropump to be beneficial to the development of the rapid, high-throughput, low-cost, and POC-PCR testing, which would have a significant impact on timely disease recognition and intervention in an emerging pandemic. Future work will include further improvement of the pump density.

## Figures and Tables

**Figure 1 micromachines-13-01565-f001:**
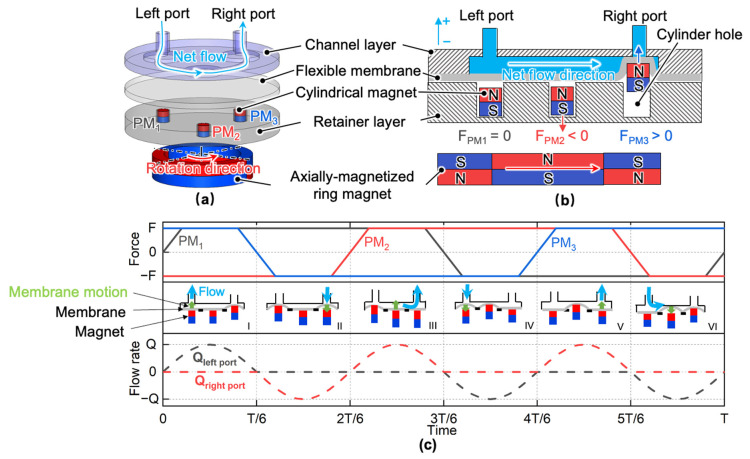
Working principle: (**a**) exploded view of the proposed micropump; (**b**) two-dimensional model of the pump at the mean radius of the ring magnet; (**c**) timing chart of vertical magnetic force on each cylindrical permanent magnet and flow rate through two ports.

**Figure 2 micromachines-13-01565-f002:**
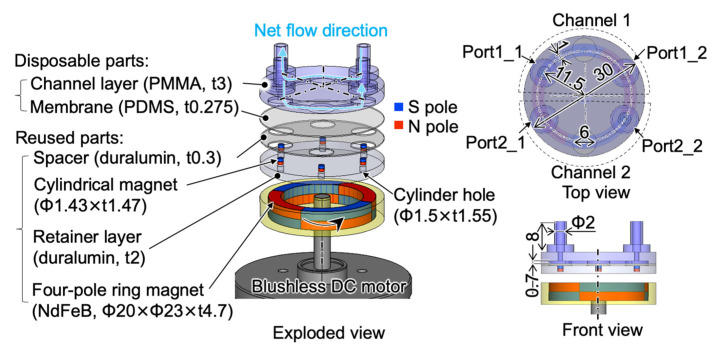
Three-dimensional model of a large-scale micropump prototype having two channels.

**Figure 3 micromachines-13-01565-f003:**
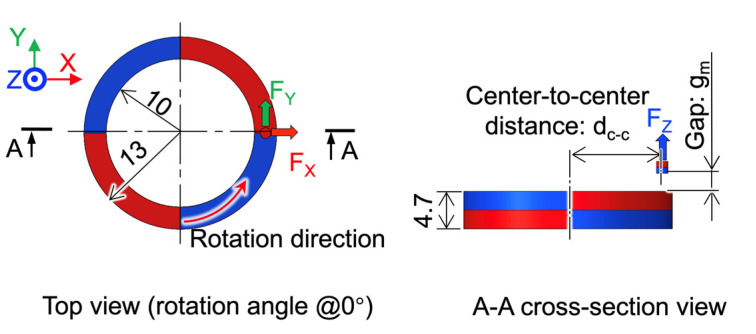
Finite element analysis model for the magnetic coupling force.

**Figure 4 micromachines-13-01565-f004:**
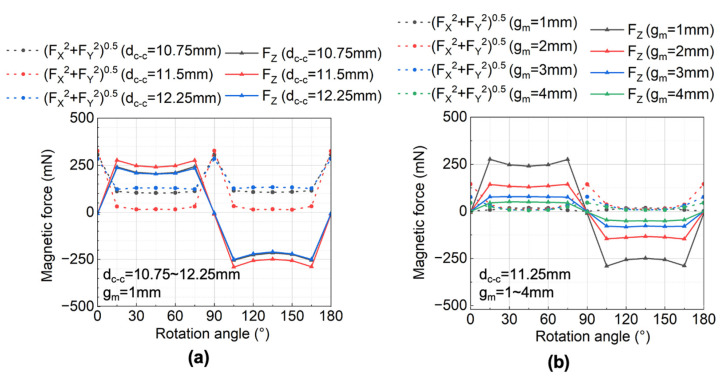
Simulated magnetic force: (**a**) effect of center-to-center distance on magnetic force; (**b**) effect of the gap between magnetic coupling on magnetic force.

**Figure 5 micromachines-13-01565-f005:**
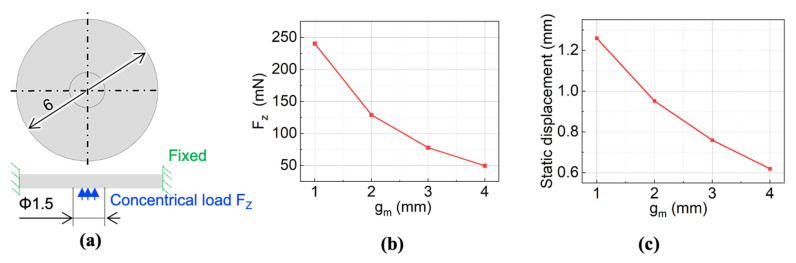
Membrane static displacement simulation: (**a**) analytical model; (**b**) simulated relationship between g_m_ and concentrical load; (**c**) simulated relationship between g_m_ and static displacement.

**Figure 6 micromachines-13-01565-f006:**
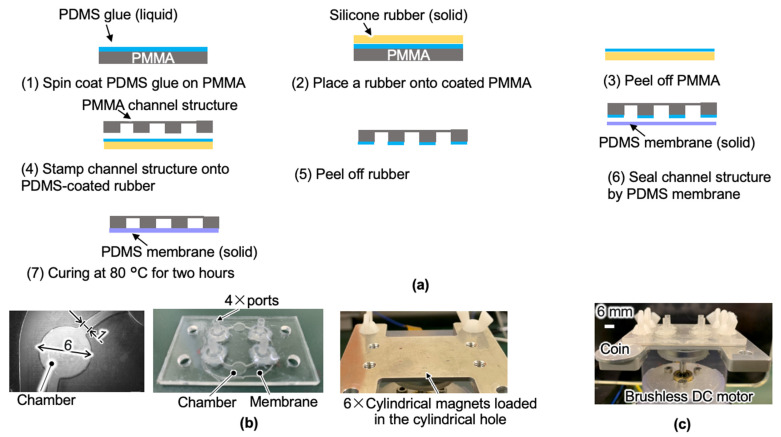
Large-scale micropump prototyping: (**a**) sealing PMMA microfluidic channel with PDMS diaphragm using PDMS liquid; (**b**) micropump parts; (**c**) micropump assembly.

**Figure 7 micromachines-13-01565-f007:**
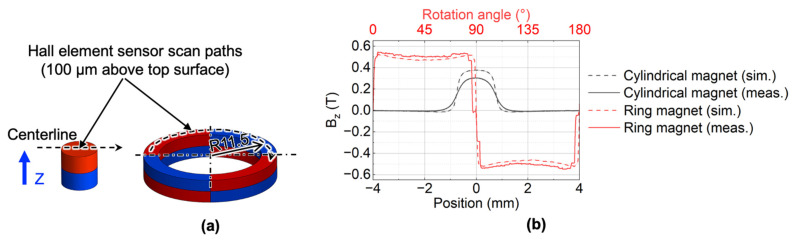
Surface magnetic flux density of the cylindrical magnet and ring magnet: (**a**) analytical models; (**b**) simulated and measured results.

**Figure 8 micromachines-13-01565-f008:**
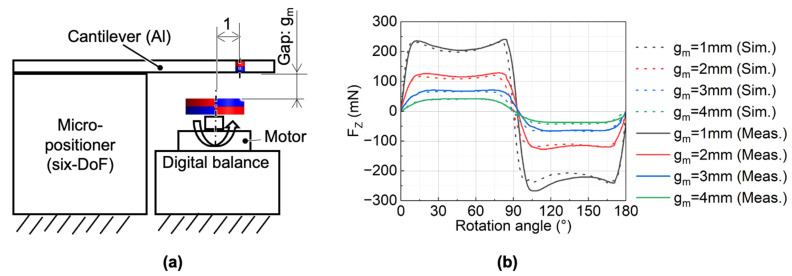
Measurement of static magnetic force: (**a**) experimental setup; (**b**) measured and simulated results (after correction) at each gap.

**Figure 9 micromachines-13-01565-f009:**
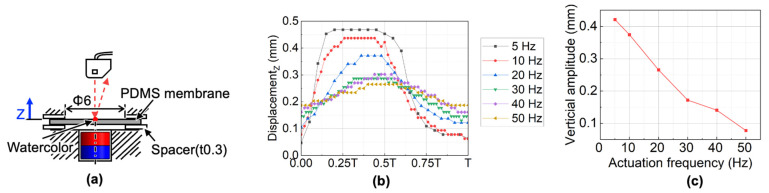
Measurement of the membrane dynamic displacement without pump flow: (**a**) measurement setup; (**b**) measured responses with and without spacer at different frequencies; (**c**) the relationship between membrane amplitude and actuation frequency.

**Figure 10 micromachines-13-01565-f010:**
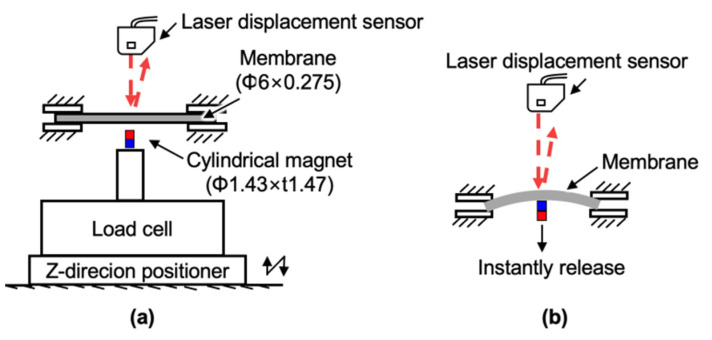
Identification of the membrane dynamic parameters: (**a**) measurement setup for membrane spring constant; (**b**) measurement setup for membrane damping coefficient; (**c**) measured spring constant; (**d**) measured transient response of the membrane.

**Figure 11 micromachines-13-01565-f011:**
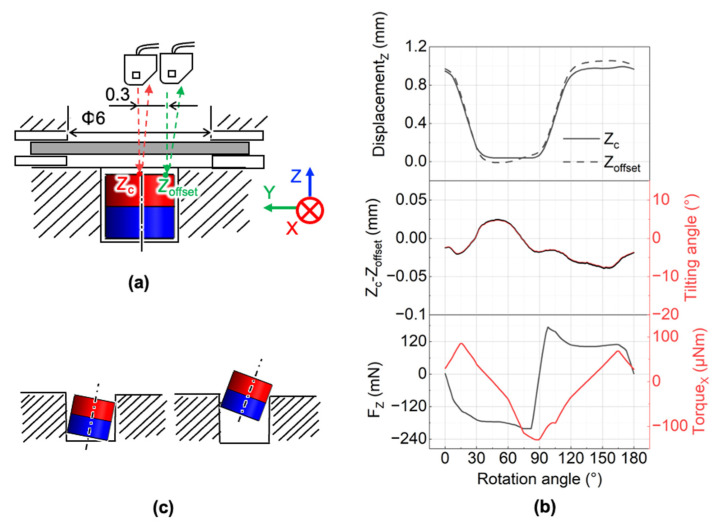
Measurement of magnet dynamic displacement and tilting angle in the air: (**a**) measurement setup; (**b**) measured tilting angle, simulated force, and simulated torque (gap: 1 mm; actuation frequency: 5 Hz); (**c**) two cases of tilting angles at different vertical displacements.

**Figure 12 micromachines-13-01565-f012:**
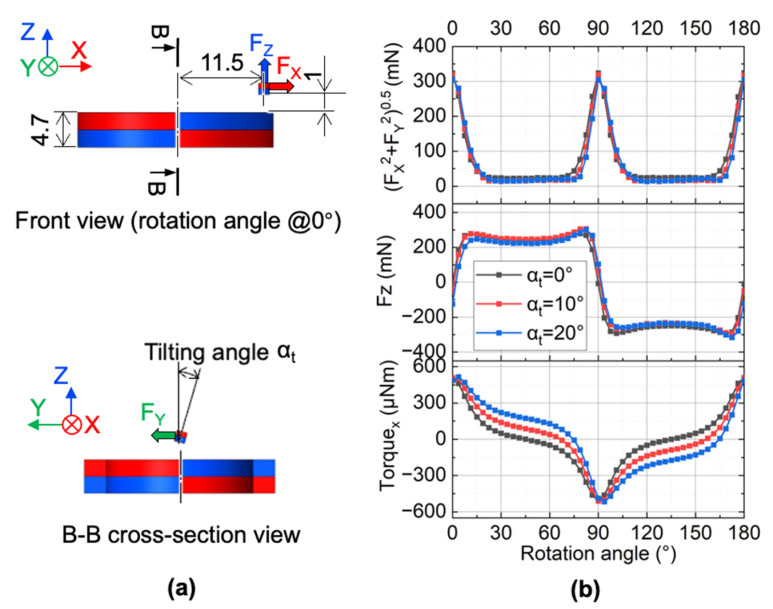
Effect of the tilting angle on the magnetic forces and torque: (**a**) analytical model; (**b**) simulation results at different tilting angles.

**Figure 13 micromachines-13-01565-f013:**
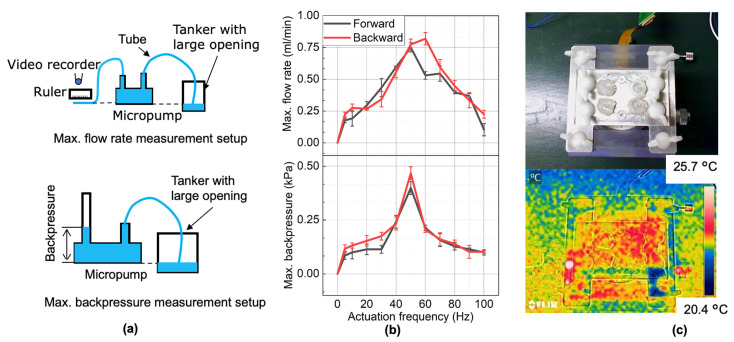
Measurement of the pump performance: (**a**) experimental setup schematic; (**b**) measurement results of maximum flow rate and maximum backpressure at different frequencies; (**c**) thermal test of the micropump after one hour of operation.

**Figure 14 micromachines-13-01565-f014:**
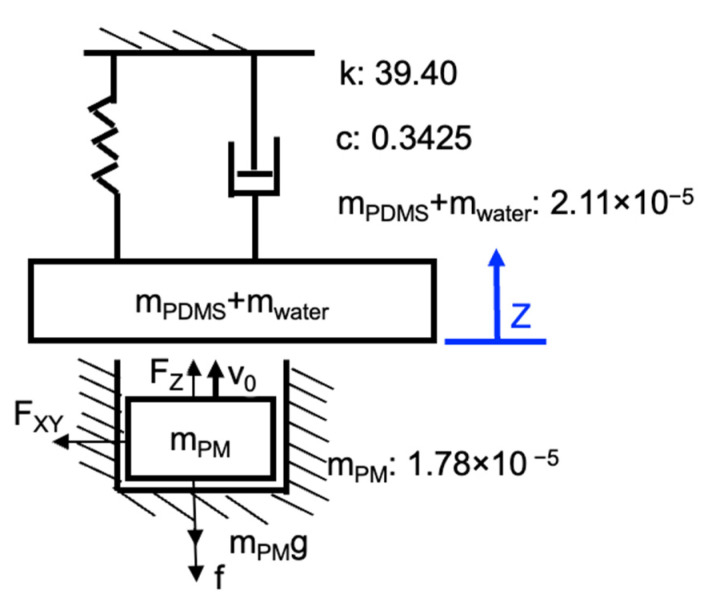
Simple lumped element model of the magnet and membrane.

**Figure 15 micromachines-13-01565-f015:**
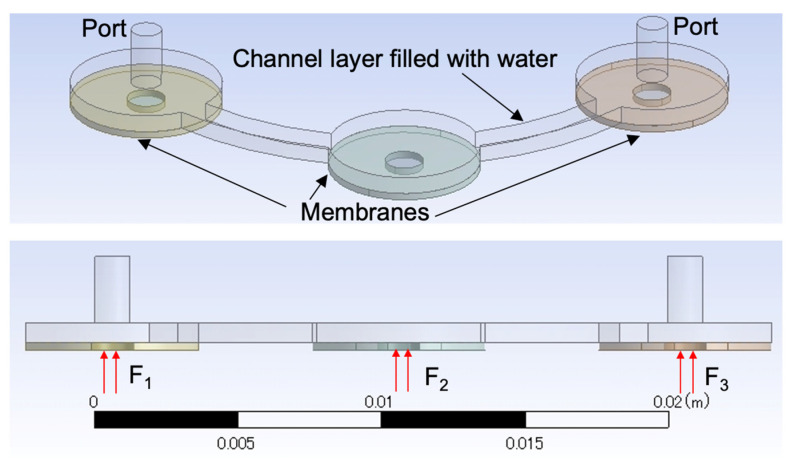
Analytical model of the proposed micropump in FSI.

**Figure 16 micromachines-13-01565-f016:**
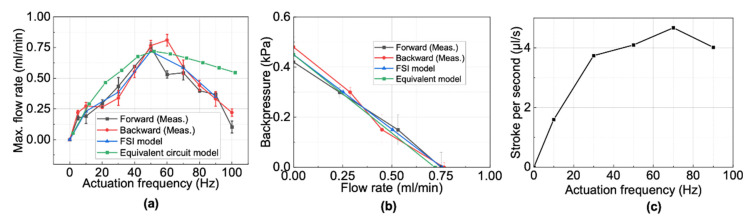
(**a**) Simulated and measured average flow rate against zero backpressure at different frequencies; (**b**) simulated and measured pump performance curve at the resonant frequency; (**c**) calculated volume stroke of one membrane per second based on lumped element model.

**Table 1 micromachines-13-01565-t001:** Bi-directional pump performance comparisons when *n* channels are assumed.

	Pump Components (For n-Channel Pump)	Disposed Components	Pump Density	Q_max_; H_max_	Voltage	Power	Stable
Unit			cm^−2^	ml/min; kPa	V	W	
Rulsi [13]	2*n* bulky coil and 2*n* magnets	2*n* magnets and fluidic part	0.16	0.068; 1.2	0.64	0.486*n*	Yes
Kim [14]	*n* motors, magnetic fluid	Magnetic fluid and fluidic part	0.30	0.0038; /	/	/	No
Du [16]	*n* motors, 3*n* steel balls and 3*n* PMs	Fluidic part	0.07	5; 10	12	/	No
Shen [18]	*n* motors, 3*n* cylindrical magnets, 6*n* arc magnets	All the component except motor	0.10	2.4; 6.6	0.7	0.02*n*	Yes
This work	One motor, 3*n* cylindrical magnets, one 2*n*-pole ring magnet	Fluidic part	0.14*n*	0.86; 0.5	4.8	0.2	Yes

## Data Availability

Not applicable.

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
