# Peer review of "A Disposable Electromagnetic Bi-Directional Micropump Utilizing a Rotating Multi-Pole Ring Magnetic Coupling"

_micromachines, 2022, doi:10.3390/mi13101565_

Round 1
Reviewer 1 Report
Good content and details of research work. However the flow and arrangement of the content is not clear. Some of the suggestions.
- Be consistent in dimension labelling. (Eg line 223)
- Section 4.2 could be improved where the title should be Modelling and measurement of….. Combine 4.2.1 and 4.2.2 to explain the details of the setup and measurement parameters for the membrane dynamic response and then follow by 4.2.3 and 4.2.4 the measurement results.
- Any study of the membrane thickness and durability? Or it is not relevant because of disposable feature?
- Section 5 on the pump modelling, any performance comparison and discussion on the results shown on Figure 13? Because Section 5 seems to be lack of connection to the other content.
Reviewer 2 Report
In the manuscript, “A Disposable Electromagnetic Bi-Directional Micropump Utilizing a Rotation Multi-pole Ring Magnetic Coupling”, Qi and co-workers have reported the fabrication of micropump to be used for specialized applications like rapid PCR. I have two minor criticisms and suggestion to improve the manuscript.
1. Authors should do a simple cost analysis of their pump to show that it’s a cheap option compared to other options.
2. Please improve the quality of your figures. All figures are little blurred.
